# Allantoin Inhibits Compound 48/80-Induced Pseudoallergic Reactions In Vitro and In Vivo

**DOI:** 10.3390/molecules27113473

**Published:** 2022-05-27

**Authors:** Ping Zhang, Yanjie Wang, Jingyu Zhang, Tie Hong

**Affiliations:** Department of Pharmacology, School of Pharmaceutical Sciences, Jilin University, Changchun 130021, China; zhangping19@mails.jlu.edu.cn (P.Z.); yjwang19@mails.jlu.edu.cn (Y.W.); jingyu20@mails.jlu.edu.cn (J.Z.)

**Keywords:** allantoin, pseudoallergy, mast cells, degranulation

## Abstract

Pseudoallergic reactions are hypersensitivity reactions mediated by an IgE-independent mechanism. Since allantoin (AT)-mediated pseudoallergy has not been studied, in this study, our objective is to investigate the anti-pseudoallergy effect of AT and its underlying mechanism. In vitro, β-hexosaminidase (β-Hex) and histamine (HIS) release assays, inflammatory cytokine assays, toluidine blue staining, and F-actin microfilament staining were used to evaluate the inhibitory effect of AT in RBL-2H3 cells stimulated with Compound 48/80 (C48/80). Western blot analysis is further performed to investigate intracellular calcium fluctuation-related signaling pathways. In vivo, Evans Blue extraction, paw swelling, and the diameter of Evans Blue extravasation were evaluated, and skin tissues are examined for histopathological examination in mice with passive cutaneous anaphylaxis (PCA) induced by C48/80. Body temperature is measured, and the levels of cytokines are further determined by ELISA kits in mice with active systemic anaphylaxis (ASA) induced by C48/80. The results show that AT dose-dependently inhibited degranulation in C48/80-stimulated RBL-2H3 cells by inhibiting β-Hex and HIS release, reducing the levels of TNF-α, IL-8, and MCP-1, inhibiting shape changes due to degranulation and disassembling the F-actin cytoskeleton. Furthermore, AT dose-dependently inhibits the phosphorylation of PLCγ and IP3R. In vivo, AT decreased Evans Blue extravasation, paw swelling, and the diameter of Evans Blue extravasation and significantly ameliorate pathological changes and mast cell degranulation in C48/80-induced PCA. Furthermore, AT help the mice recover from the C48/80-induced decrease in body temperature and decreased the levels of cytokines in C48/80-treated ASA mice. Our results indicate that allantoin inhibits compound 48/80-induced pseudoallergic reactions. AT has the potential to be used in IgE-independent anti-allergic and anti-inflammatory therapies.

## 1. Introduction

Allergic diseases such as anaphylaxis, asthma, airway hyperresponsiveness, eczema, and eosinophilic disorders affect nearly 25% of people in developed countries [1]. Although allergic diseases are usually referred to as immunoglobulin E (IgE)-type hypersensitivity, non-IgE-mediated allergic reactions, also known as pseudoallergic reactions, are also important types of allergic diseases [2,3].

Mast cells (MCs) are crucial effectors in allergic disorders [4]. MRGPRX2 can be directly activated by many stimuli, including C48/80, substance P, and cathelicidin [5,6]. Previous studies have pointed out that MRGPRX2 expressed on human MCs is directly activated by C48/80, causing phosphorylation of phospholipase Cγ (PLCγ) [7]. PLC-γ and IP3R are key proteins involved in the regulation of intracellular calcium concentration and the degranulation of MCs through the Ca^2+^/PLC/IP3 pathway [8,9]. Activated MCs also secrete many biologically active mediators, which include β-Hex, HIS, and pro-inflammatory cytokines [10,11] (such as TNF-α, IL-8, and MCP-1).

Allantoin (Figure 1) possesses physiological functions, such as promoting cell growth [12], accelerating wound healing [13], and softening keratin [14]. It is a good healing agent for skin wounds. It can be used to relieve and treat dry skin, scaly skin diseases, skin ulcers, and inflammation [15], but until now, there has been no study of the protective effect of AT against pseudoallergic reactions or its mechanism. In the present study, the effects and mechanism of AT in C48/80-induced pseudoallergic reactions were reported.

## 2. Results

### 2.1. Effects of AT on the Release of β-Hex, HIS, and Cytokines by C48/80-Stimulated RBL-2H3 Cells

In this study, we first investigated the effect of AT on RBL-2H3 cell viability. As shown in Figure 1A, stimulation with C48/80 did not reduce RBL-2H3 cell viability. MC activation appeared as degranulation, accompanied by the secretion of many inflammatory mediators, which are essential for allergic inflammation. The release of preformed mediators, β-Hex, HIS, and cytokines in the supernatant was evaluated. The results indicated that β-Hex and HIS were inhibited in a dose-dependent manner by treatment with AT (Figure 1B,C). Similarly, AT dose-dependently reduced the release of TNF-α, IL-8, and MCP-1 in the supernatant (Figure 1D–F) compared with the C48/80 group.

### 2.2. Effect of AT on the Release of Granules by C48/80-Stimulated RBL-2H3 Cells

As shown in Figure 2, the normal RBL-2H3 cells (Figure 2A) were spindle-shaped and had purple granules stored in the cells. However, the shape of the C48/80-stimulated RBL-2H3 cells (Figure 2B) was irregular, and purple granules were released outside the cell. Treatment with AT (Figure 2C,D) markedly inhibited the morphological changes due to degranulation.

### 2.3. Effect of AT on Membrane Ruffling in C48/80-Stimulated RBL-2H3 Cells

As shown in Figure 3, the normal RBL-2H3 cells were spindle-shaped with a uniform distribution of F-actin at the cell periphery (Figure 3A). The shapes of C48/80-stimulated RBL-2H3 cells became round or elliptical due to the disassembly of the F-actin cytoskeleton (Figure 3B). Treatment with AT inhibited the shape change and disassembly of the F-actin cytoskeleton (Figure 3C,D).

### 2.4. Effect of AT on the PLCγ/IP3R Signaling Pathway in C48/80-Stimulated RBL-2H3 Cells

Western blotting showed that the phosphorylation levels of PLCγ and IP3R were significantly enhanced in the model group compared with the control group. However, these effects were dose-dependently downregulated by AT treatment (Figure 4A,B).

### 2.5. Effect of AT on C48/80-Induced Local Cutaneous Inflammation in PCA

As shown in Figure 5A and Figure 6A, there was no Evans Blue extravasation in the paws and back skin of the mice in the control group, but Evans Blue extravasation was the highest in the model group. After pretreatment with AT, Evans Blue extravasation decreased in a dose-dependent manner. Paw swelling and diameter also showed similar reduced results after treatment with AT in a dose-dependent manner (Figure 5B,C and Figure 6B,C).

C48/80-induced telangiectasis (capillary expansion and an increased number of erythrocytes) was constricted after pretreatment with AT (Figure 5D and Figure 6D). The degranulated MCs located in the paw and back skins were significantly decreased by AT via toluidine blue staining (Figure 5E and Figure 6E), which was used to identify MCs for their ability to combine with MC granules.

### 2.6. Effect of AT on C48/80-Induced Active Systemic Anaphylaxis

In the analysis of systemic anaphylaxis, compared with the control group, the body temperature of the model group decreased significantly. After administration of AT, the body temperature increased significantly (Figure 7A). Furthermore, the C48/80-induced serum concentrations of HIS (Figure 7B), TNF-α (Figure 7C), and IL-8 (Figure 7D) were reduced by AT pretreatment.

## 3. Discussion

In this study, the effects of AT on pseudoallergic reactions were evaluated in vivo and in vitro. The results showed that AT dose-dependently inhibited degranulation in C48/80-stimulated RBL-2H3 cells, such as inhibiting β-Hex and HIS release, reducing the levels of TNF-α, IL-8, and MCP-1, and inhibiting shape changes due to degranulation or disassembly of the F-actin cytoskeleton. Furthermore, AT dose-dependently reduced Ca^2+^ influx while inhibiting the phosphorylation of PLCγ and IP3R.

In passive cutaneous anaphylaxis, telangiectasis was constricted after pretreatment with AT. Degranulated MCs located in the paw and back skins were significantly decreased by AT. In the analysis of systemic anaphylaxis, the body temperature recovered and the concentrations of HIS, TNF-α, and IL-8 in serum were reduced by AT pretreatment.

RBL-2H3 cells with a variety of biological properties of mast cells are known as a classic model to study the degranulation response [16]. C48/80 was first reported in 1951 as an active histamine-releasing agent [17] that can cause redness, itching, and itching of human skin [18]. C48/80 was later used as a classical mast cell activator for IgE-independent G proteins and is the most commonly used activation method in pseudoallergy studies [19]. β-Hex is a biomarker for MC degranulation [20]. HIS secreted by mast cells induces vasodilation, edema, pruritus, and hypothermia [21]. IL-8 is a chemoattractant substance for the recruitment of eosinophils and neutrophils [22]. TNF-α and MCP-1 are also strongly associated with the progression of inflammatory cytokines because they promote the pathogenesis of acute and chronic inflammatory diseases [23,24]. In the present study, β-Hex and HIS release assays and inflammatory cytokine assays were used to evaluate the inhibitory effect of AT in RBL-2H3 cells induced by Compound 48/80. Our experimental results showed that β-Hex, HIS, TNF-α, IL-8, and MCP-1 were decreased.

Toluidine blue staining readily identifies mast cell metachromatic granules against a pale blue background [25]. Therefore, we used toluidine blue staining to confirm that pretreatment with AT preserved cell morphology, including the recovery of an elongated shape and the release of fewer granules. Rearrangements of the F-actin cytoskeleton, a key step of the degranulation process, facilitate granule fusion with the plasma membrane [26]. Our results showed that AT also suppresses the rearrangement of the F-actin cytoskeleton.

The key factor that triggers MC degranulation is the increase in the intracellular Ca^2+^ concentration [27]. It is generally believed that the degranulation induced by immune stimuli depends on the influx of extracellular Ca^2+^ into the cell [22], which increases the intracellular Ca^2+^ concentration [28]. At the same time, inositol triphosphate (IP3) can also be generated to release Ca^2+^ stored in the endoplasmic reticulum [29]. Another possibility is that it activates phospholipase γ (PLCγ) directly or indirectly [30], thereby activating protein kinase C and opening the intracellular Ca^2+^ storehouse [31], which also triggers the degranulation of MCs. Our results showed that AT can inhibit the phosphorylation of PLCγ and IP3R proteins during MC degranulation, suggesting that AT can inhibit the PLCγ/IP3R signaling pathways.

In human subjects, most MCs are found in connective tissues, such as the skin [32]. Therefore, we used the PCA model to measure the AT effects in vivo. Our experimental results showed that there was significant Evans Blue extravasation in the induced area of the model group, with the largest range, the darkest color, and the highest degree of swelling, indicating that the local skin of the mice had allergic reactions. Compared with the model group, the diameter of Evans Blue extravasation in the corresponding part of the AT group decreased, the color became lighter, the degree of swelling decreased, and the dosage of AT was positively correlated with the anti-pseudoallergy effect. The sensitized areas were stained with H&E and toluidine blue, which further proved that the diminished allergy symptoms are positively associated with the inhibitory effects of AT on MC activation, especially the release of histamine, which is beneficial for vasodilation and increased vascular permeability [33]. In further active systemic hypersensitivity experiments, AT helped the mice recover from the C48/80-induced decrease in body temperature, which might be due to the inhibition of HIS release in serum. The levels of some cytokines are elevated in serum during anaphylaxis in humans and are associated with severe clinical symptoms [34]. The results showed that the levels of TNF-α, IL-8, and MCP-1 were dramatically decreased following AT administration.

In summary, AT exerted a potent inhibitory effect on pseudoallergic reactions both in vivo and in vitro, which was related to the suppression of the Ca^2+^/PLCγ/IP3R signaling pathways, thus indicating that AT has the potential to be used in IgE-independent anti-allergic and anti-inflammatory therapies.

## 4. Materials and Methods

### 4.1. Drugs and Reagents

AT (purity ≥ 98%), compound 48/80 (C2313), 4-Nitrophenyl *N*-acetyl-β-D-glucosaminide, and Triton X-100 were from Sigma-Aldrich (St. Louis, MO, USA). Fluo-4 AM was from Beyotime Biotech Inc. (Beijing, China). Tyrode’s buffer was prepared fresh on the day of use (6.954 g/L NaCl, 0.353 g/L KCl, 0.282 g/L CaCl_2_, 0.143 g/L MgSO_4_, 0.162 g/L KH_2_PO_4_, 2.383 g/L HEPES, 0.991 g/L glucose, and 1 g/L BSA, pH = 7.4).

### 4.2. Cell Culture

RBL-2H3 cells were obtained from the National Infrastructure of Cell Line Resource (Shanghai, China) and maintained in minimum essential medium (MEM) with 15% fetal bovine serum (FBS), 100 μg/mL streptomycin, 100 U/mL penicillin, 1.5 mg/mL sodium bicarbonate, and 110 μg/mL sodium pyruvate at 37 °C in a humidified incubator with 5% CO_2_.

### 4.3. Animals

Female BALB/c mice weighing 20–22 g were purchased from Liaoning Changsheng Biotechnology Co., Ltd., Shenyang, China (license number SCXK (Liao) 2020-0001). Before the experiments, all mice were kept under standard conditions with a regular 12 h light/dark cycle and free access to food and water for 3 days. All animal experiments were conducted under a protocol that was approved by the Institutional Animal Care and Use Committee of Jilin University.

### 4.4. Cell Viability Assay

The MTT [3-(3,5)-dimethylthiazol-2,5-diphenyltetrazolium bromide] assay was used to determine cell viability. RBL-2H3 cells were seeded into a 96-well plate and incubated at 37 °C overnight. Then, the cells were treated with AT different concentrations (0, 3.75, 7.5, 15, 30, 60, and 120 μM) for 30 min. Next, the cells were treated with 10% MTT for 4 h. The formazan crystals produced in the cells were dissolved with DMSO added to each well. A microplate reader (Biotek Instruments, Inc., Winusky, VT, USA) was used to measure relative cell viability at 490 nm absorbance.

### 4.5. β-Hex and HIS Release Assay

RBL-2H3 cells were seeded into a 48-well plate at 37 °C overnight and then treated with different concentrations (between 30 and 60 μM) of AT. After 30 min, the cells were stimulated with C48/80 for 30 min. The cells were incubated in an ice bath for 10 min to stop the reaction and centrifuged at 300× *g* for 10 min at 4 °C. The supernatant was mixed with substrate (1 mM p-nitrophenyl-*N*-acetyl-β-D-glucosaminide in 0.1 M sodium citrate buffer, pH 1.5), and the fixative solution was incubated for 1.5 h at 37 ℃. The reaction was terminated by the addition of buffer (0.1 M Na_2_CO_3_/NaHCO_3_, pH 10.0).

For the HIS release assay, the supernatants were collected, and the content of histamine was measured according to the protocol of the enzyme-linked immunosorbent assay (ELISA) kit (Shanghai FANKEL Industrial Co., Ltd., Shanghai, China). A microplate reader (Biotek Instruments, Inc., Winusky, VT, USA) was used to measure the absorbance at 405 nm.

### 4.6. Inflammatory Cytokine Assay

After stimulation with C48/80, the cell culture supernatant was collected to measure the levels of IL-8, MCP-1, and TNF-α according to the protocols of ELISA kits (Shanghai FANKEL Industrial Co., Ltd., Shanghai, China).

### 4.7. Toluidine Blue Staining

RBL-2H3 cells stimulated with C48/80 for 30 min were washed with phosphate-buffered saline (PBS) and then incubated with 4% paraformaldehyde/PBS for 30 min at room temperature (RT). The mixture was discarded, and fixed cells were stained with toluidine blue dye for 30 min. Images of the stained cells were then examined and captured using an inverted microscope.

### 4.8. F-Actin Microfilament Staining

RBL-2H3 cells stimulated with C48/80 for 30 min (in 4-well chamber slides) were washed with PBS and fixed using 4% paraformaldehyde/PBS for 1 h and 30 min. The fixed cells were washed with PBS and then permeabilized with 0.1% Triton X-100/PBS for 3 min. The permeated cells were washed with PBS and then stained using Alexa Fluor 488-phalloidin for 30 min. Finally, F-actin fibers were examined using a Leica DM2500 Microscope equipped with excitation (490 nm) and emission (520 nm) filters.

### 4.9. Western Blot

The protein concentrations were measured using a BCA protein assay kit (Beyotime, Beijing, China). The proteins were then separated using 10% SDS–PAGE and transferred to PVDF membranes. After blocking in 5% skim milk for 2 h, the membranes were incubated with primary antibodies at 4 °C overnight and then incubated with secondary antibodies for 1 h at RT. Enhanced chemiluminescence reagent was used for signal visualization. Anti-PLCγ (D9H10, PLCγ, 1:1000), anti-phospho-PLCγ (D25A9, P-PLCγ, 1:1000), anti-IP3R (D53A5, IP3R, 1:1000), anti-phospho-IP3R (D10E3, P-IP3R, 1:1000), and anti-GAPDH (D16H11, GAPDH, 1:2000) antibodies were purchased from Cell Signaling Technology (Boston, MA, USA). Goat anti-mouse IgG (32160702, 1:4000) and goat anti-rabbit IgG (32160702, 1:4000) secondary antibodies were purchased from Life Science (Santa Cruz, CA, USA).

### 4.10. Passive Cutaneous Anaphylaxis

Female BALB/c mice were randomly divided into the following 4 groups: the control, model, and AT groups (25 mg/kg and 50 mg/kg). Mice were orally administered AT (25 mg/kg, 50 mg/kg) for 30 min before anesthetization with an intraperitoneal injection of 50 mg/kg pentobarbital sodium, followed by intravenous (tail vein) injection with 0.2 mL 0.4% Evans Blue dye in saline. The control and model groups were given only the corresponding volume of saline. Five minutes after the end of administration, 5 μL of C48/80 (30 μg/mL) was injected intracutaneously into the left paw and back skin of the model and AT groups for sensitization. The mice in the control group were given only the corresponding volume of saline. Thirty minutes later, the mice were sacrificed, and the paws and back skin were photographed for recording. After that, a paw of the same size and the pigmented area on the back were taken around the injection site; the thickness of the paw and the diameter of the blue spot on the back of the mouse were measured with a Vernier caliper, and the paw swelling was calculated. The paws and back skin were dried at 60 °C for 24 h and then placed in 1 mL of acetone-physiological saline and placed in a water bath at 85 °C for 2 h. After centrifugation at 150× *g* for 20 min, the supernatant was collected to detect the absorbance at 620 nm.

### 4.11. Histological Analysis

Mouse tissues were fixed with 4% paraformaldehyde for 24 h and then embedded in paraffin. After being sectioned, the tissue was used for hematoxylin-eosin (H&E) staining and toluidine blue staining, followed by optical microscopic inspection.

### 4.12. Active Systemic Anaphylaxis

Female BALB/c mice were randomly divided into the following 4 groups: the control, model, and AT groups (25 mg/kg, 50 mg/kg) (intragastric administration). Thirty minutes after gavage, C48/80 (0.3 mg/kg) was injected through the tail vein, and body temperature was recorded using a biological function experimental system, in which a probe was inserted into the anus of each mouse for 30 min.

### 4.13. Enzyme-Linked Immunosorbent Assay (ELISA)

In active systemic anaphylaxis, serum levels of HIS (sensitivity: 0.2 μg/mL) and MCP-1 (sensitivity: 3.75 pg/mL), IL-8 (sensitivity: 2 pg/mL), and TNF-α (sensitivity: 6.25 pg/mL) were measured using specific ELISA kits (Shanghai FANKEL Industrial Co., Ltd., Shanghai, China) according to the manufacturer’s instructions. Then, the optical density was spectrophotometrically measured at 450 nm using a microplate reader (Biotek Instruments, Inc., Winusky, VT, USA).

### 4.14. Statistical Analysis

The SPSS 20.0 statistical software package was used to perform all statistical analyses. All experimental data are expressed as the mean ± standard deviation (S.D.). Comparisons between groups were made by one-way analysis of variance (ANOVA). A value of *p* < 0.05 was considered significant.

## Data Availability

The data presented in this study are available on request from the corresponding author.

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
