# Peer review of "Allantoin Inhibits Compound 48/80-Induced Pseudoallergic Reactions In Vitro and In Vivo"

_molecules, 2022, doi:10.3390/molecules27113473_

Round 1
Reviewer 1 Report
In the manuscript by Zhang et al. "Allantoin inhibits Compound 48/80-induced pseudoallergic reactions in vitro and in vivo" submitted to Molecules, the authors present a study showing the effects and mechanism of allantoin (AT) in C48/80-induced pseudoallergic reactions. The effect of AT on some aspects of mast cell (MC) activity was analyzed. Also, the effects of AT on pseudoallergic reactions were evaluated in vivo. Generally, the article is of interest, however, there is a lack of many important issues to confirm the presented findings, and therefore manuscript requires significant revision. I believe that Reviewer suggestions are important for improving this paper. So I recommend a major revision.
Major points:
- Since in vivo studies were performed using BALB/c mice, is there a reason why the RBL-2H3 cell line was used rather than more easily obtained mouse primary mast cells? It is widely known that the cellular models representing MCs, such as immortalized MC lines or MCs cultured and maturated in vitro in "artificial" laboratory conditions, which never accurately reflect physiological conditions, are far away from MCs found in vivo regarding biological properties.
- Figure 1: authors use AT of 3.75 to 120 µM. Are these concentrations reasonable to investigate the anti-pseudoallergy effects? What was the time of stimulation for cytokine secretion? How these times were justified?
- Cell viability assay: The vehicle control should be added.
- How did the authors choose the cytokines? Is it based on literature or previous results? This issue should be discussed.
- Authors claim that they analyze the effects of AT on the Ca2+ concentration in C48/80-stimulated RBL-2H3 cells. However, the results (images) from intracellular calcium measurement are not quantitative and they are illegible. Therefore authors should use another assay to obtain dynamics of intracellular Ca2+ concentration after AT stimulation (e.g. using fluorescent ratiometric dye fura-2 AM).
- Figure legends: Authors state that ‘The data are presented as the mean ± S.D.’ However, it is not a clear statement. What it does mean? What number of experiments were performed? What number of samples were analyzed?
- Figure 2 is not readable in the current version. A part of each image of this figure should be magnified in order to better visualize MC granules.
- The same remark as above mentioned refers to Figure 3. A part of each image of this figure should be magnified. Moreover, is there a possibility to provide this figure in a better resolution?
- Figure 5A: I strongly suggest showing a picture/scan with the entire blots.
- Can authors explain why they conduct their studies using only females animals?
- Figure 6: Is there a possibility to provide this figure in a better resolution?
- 'Western Blotting' section: Specifications of primary and secondary antibodies should be given. There is also a lack of information regarding antibodies dilution.
- The sensitivity of ELISA assays should be provided.
- The entire article needs editorial corrections, e.g. section 11. Passive cutaneous anaphylaxis – different font types.
Author Response
Dear reviewer,
Please see the attachment.
Your sincerely,
Tie Hong.

Reviewer 2 Report
It is interesting to study Allantoin in Compound 48/80-induced pseudoallergy.
Some suggestions:
- Some more background about Allantoin should be added in the introduction, such as the structure of allantonin.
- For the membrane ruffling in C48/80-stimulated RBL-2H3 cells, the power and the resolution of the figures are too low. High power and high resolution image of the cells should be used in order to demonstrate what the authors claimed.
- Calcium signaling is a dynamic event over time. Compound 4880- induced calcium influx is quite different from antigen stimulation in mast cells. Calcium is not likely to be accumulated after 48/80 stimulation. The drug may transiently affect the magnitude of the 4880 calcium influx (a few seconds) and the authors should clearly document the conditions in taking the pictures. It is more convincing to show a time course of the calcium change and quantify the change for an objective comparison.
- The resolution of figure 6 and 7 (D and E) is too low for scientific presentation.
Author Response

(The authors gave the same response as above.)

Round 2
Reviewer 1 Report
The Authors have addressed all of my concerns with the original manuscript.
Author Response
Dear reviewer,
Thanks for your time review this manuscript again.
Your sincerely,
Tie Hong.
Reviewer 2 Report
For the calcium study, the images for a single time frame are not sufficient for an objective conclusion.
I have come across a very similar published study for pseudo allergy:
Biochemical PharmacologyVolume 148, February 2018, Pages 147-154
Saikosaponin A inhibits compound 48/80-induced pseudo-allergy via the Mrgprx2 pathway in vitro and in vivo
The method for this study is more objective to demonstrate the calcium changes in mast cells. I suggest the authors repeat the study as the paper or at least show 5-6 time frames from 0 -5 minutes of the calcium changes pictures for their study.
Compound 48/80 is a calcium-independent mast cell stimulator. In my opinion, it is not important to demonstrate the inhibitory activity of Allantoin is calcium-dependent or not if you use compound 48/80 for your study. The story might be better if the calcium results (present form) are taken away.
Author Response
Dear reviewer,
Thanks for your time and comment, we completely agree with that for the calcium study, the images for a single time frame are not sufficient for an objective conclusion. We already taken away pictures (calcium result) from our manuscript.
Your sincerely,
Tie Hong